Elemental pollution and risk assessment of soils and Gundelia tournefortii in a multi-sector industrial zone with a history of agricultural use

Özuysal Ayşenur aysenur.bolukbas@deu.edu.tr 1
Fadaeivash Fariborz 2
Akıncı Görkem 1 3
1 Dokuz Eylul University, Department of Environmental Engineering , Buca , İzmir , Turkey
2 Dokuz Eylul University, The Graduate School of Natural and Applied Sciences, Environmental Engineering , Buca , İzmir , Turkey
3 Dokuz Eylul University, Science and Technology Application and Research Center , Buca , İzmir , Turkey
Serim Ahmet Tansel
Electronic publication date: 2025 Nov 24
Publication date: 2025
Volume: 13
Electronic Location ID: e20374
Received 2024 Dec 20; Accepted 2025 Oct 20
Copyright: ©2025 Özuysal et al.
Copyright year: 2025
Copyright holder: Özuysal et al.
License: This is an open access article distributed under the terms of the Creative Commons Attribution License, which permits unrestricted use, distribution, reproduction and adaptation in any medium and for any purpose provided that it is properly attributed. For attribution, the original author(s), title, publication source (PeerJ) and either DOI or URL of the article must be cited.
License URL: https://creativecommons.org/licenses/by/4.0/

Keywords: Gundelia tournefortii, Heavy metals, Industrial emissions, PCA, Soil

Funding: The authors received no funding for this work.

==============================
The study provides new insights into elemental enrichments in soil and Gundelia tournefortii (GT) parts in a complex land use region where intensive agricultural activities were carried out in the past and 343 industrial facilities from various sectors have been operating for the last 32 years. The levels of crustal elements (aluminum (Al), iron (Fe), potassium (K), sodium (Na), titanium (T)), heavy metals (cadmium (Cd), chromium (Cr), copper (Cu), manganese (Mn), nickel (Ni), lead (Pb), zinc (Zn)), sulfer (S) and phosphorus (P) were determined in the sample matrices, their possible sources and the health risks associated with their human consumption were investigated. Significant enrichments in Cr, Ni, Pb, and Zn were observed in the soils, with the exceptionally high Pb enrichment (EF: 81.34) being noteworthy. The soil pollution index values (PI:2.06–6.82) confirm significant anthropogenic contamination. Bioconcentration factors (BCF) for Cu, K, Na, Mg, P and S were found to be >1 in all roots and stems, Zn showed high accumulation in all roots and most stems, while Cd, Cr, Pb and Mn accumulated in a more limited number of samples. Principal component analysis (PCA) showed that the elements found in the geochemical composition of the region and those representing agricultural chemicals used throughout the long agricultural history overlap and form clusters that cannot be fully separated, while the analysis of the datasets from GT parts yielded similar results. However, the effects of industrial emissions and solid fuel use were clearly evident in the GT root and stem samples. High Nemerow Compound Pollution Index values (NCPIs) indicated contamination in plant parts. Estimated daily intake (EDI) values for Cu and Mn exceeded the tolerable upper intake level (TUIL) for children in many root and stem samples, while EDI values for Cr, Fe, and Zn also exceeded the limit value in several samples. Risk assessments for non-carcinogenic effects showed that numerous samples surpassed the safety limit for children as a result of elevated levels of Cr, Cu, Pb, Zn, Fe and Mn. Estimates for carcinogenic risk (CR) suggested that Cd and Cr in the majority of samples, along with Ni in some samples, could pose a lifetime cancer risk for children. The results reveal that geogenic influences, as well as pressures from past agricultural production and current industrial and fossil fuel-related pressures, are evident on the region’s soils and GT crops. The accumulation of toxic elements in edible plant parts poses a risk to food security, necessitating detailed risk assessments. The findings provide a scientific basis for land-use planning and agricultural management, emphasizing the importance of effective emissions monitoring, agricultural production in areas away from polluting sources, and implementing stricter land-use policies for protecting the environment and public health.

Introduction

Wild edible plants (WEPs) are an important part of the diet in rural areas and contribute to the cultural and genetic heritage of various regions (Grivetti & Ogle, 2000; Ranfa et al., 2014). WEPs are critical for global food security, nutrition and sustainable food systems as they are rich in vitamins A, E and C; minerals such as iron, calcium and magnesium; dietary fiber and many phytochemicals with antioxidant properties such as polyphenols. This nutritional diversity and their ability to readily adapt to different environmental conditions make WEPs a resilient and beneficial addition to human nutrition (Pinela, Carvalho & Ferreira, 2017; Sarkar, Walker-Swaney & Shetty, 2020; Turner et al., 2011). They also contribute to soil health and natural pest control, increasing the resilience of local agroecosystems and strengthening the adaptive capacity of food systems to climate change (Bhatti et al., 2022; Ray, Ray & Sreevidya, 2020).

WEPs are traditionally collected from nature and consumed by local people. If integrated into agriculture and cultivated, they provide an additional source of income for farmers and rural communities and contribute to ecologically sustainable development at local and global scales (Bvenura & Sivakumar, 2017; Pinela, Carvalho & Ferreira, 2017).

Gundelia tournefortii (GT) is a versatile plant widely used in cousines and traditional medicine in the Middle East and Eastern Mediterranean. It is used in the treatment of conditions such as pain, liver diseases, kidney stones, and inflammation; and it has been reported to be particularly effective in liver diseases and iron overload conditions (Jamshidzadeh et al., 2005; Khanzadeh et al., 2012; Bati et al., 2023; Mansi, Tabaza & Aburjai, 2020). Studies have shown that GT extract has the potential to lower cholesterol, body mass index, and blood sugar (Farhang, Vahabi & Allafchian, 2016; Mansi, Tabaza & Aburjai, 2020; Haghi, Hatami & Arshi, 2011; Hajizadeh-Sharafabad et al., 2016; Keskin, Kaya & Keskin, 2021). It is also emphasized that it plays a role in the prevention and treatment of many diseases due to being a food source rich in healthy fats, antioxidants, and detoxification properties (Azeez & Kheder, 2012; Çoruh et al., 2007; Goorani et al., 2018; Bati et al., 2023).

The “Farm to Fork” and “Biodiversity” strategies under the European Green Deal aim to make EU food systems more sustainable (European Commission, 2020). The Food and Agriculture Organization (FAO) promotes climate-resilient, sustainable, and inclusive agricultural food systems with the aim of reducing the effects of climate change on agriculture and strengthening the supply of healthy and safe food (Bizzo, Fabbri & Gajdzinska, 2023; Food and Agriculture Organization of the United Nations, 2021; Food and Agriculture Organization of the United Nations, 2023a). The European Union also aims to ensure that 75% of agricultural soils, which are critical for biodiversity, food security, and social resilience, are healthy for food, people, nature, and the climate by 2030 through the “European Soil Pact” mission defined under the Horizon Europe Framework Program (European Commission, 2023a; European Commission, 2023b; Veerman, 2023). Due to the impacts of climate change and the need to develop new strategies for food security and soil health, WEPs are increasingly being discussed as good and healthy alternative food sources.

However, WEPs growing in industrial and urban areas can pose risks to human and animal health by absorbing atmospheric pollutants (in the form of particulate matter or gases) or accumulating toxic compounds in the soil (Edelstein & Ben-Hur, 2018; Ranfa et al., 2014). These pollutants, ingested through the consumption of WEPs, can disrupt organ function, affect hormonal balance, cause DNA damage and oxidative stress, leading to cancer, heart disease, and neurodegenerative disorders (Sadee, 2022; Mohsenzadeh & Rad, 2012). As toxic substances accumulate throughout the food chain, humans are exposed to higher levels, increasing health risks.

Heavy metals are pollutants that occur naturally in the environment but significantly increase their accumulation in soil, water, and air through human activities such as industry, mining, agriculture, transportation, and waste disposal, leading to environmental pollution and health risks (Micó et al., 2006; Wuana & Okieimen, 2011). Table S1 presents the main heavy metals found in different industrial emissions. These metals can degrade soil properties, reduce fertility, inhibit plant growth and negatively affect the food chain by reducing agricultural yields (Chen, Wong & Zhang, 2023; Facchinelli, Sacchi & Mallen, 2001). In a study by Akinci & Ozuysal (2022), it was shown that anthropogenic heavy metal exposure during growth in vegetables even affects the metal load on municipal biodegradable wastes. In humans, these metals may cause harmful effects on neurological, respiratory, cardiovascular and reproductive systems, especially for children and pregnant women.

Therefore, heavy metals are one of the most intensively investigated pollutants in the field of environmental health. Heavy metal accumulation in agricultural crops and WEPs has been previously investigated (Table S2) (Jia et al., 2010; Rai et al., 2019; Sharma, Agrawal & Marshall, 2009). In most studies, soil cadmium (Cd) values are above the average for the Earth’s crust (Table S3), chromium (Cr) levels are high around chemical industries and waste sites, while copper (Cu), manganese (Mn), lead (Pb), and zinc (Zn) levels are generally high in soils near industries (Brunetti et al., 2009; Yaylali-Abanuz, 2011; Zhang et al., 2021).

Although there are only a limited number of studies on GT plants, these studies have shown that the plant has a high accumulation capacity for heavy metals such as Cd, Cu, Fe, Mn, nickel (Ni), Pb, and Zn, although this capacity varies depending on the environmental conditions in which the plant grows (Table S4) (Chehregani, Mohsenzade & Vaezi, 2009; Ghaderian & Ghotbi Ravandi, 2012; Jalali & Fakhri, 2021; Moameri et al., 2017; Mohsenzadeh & Rad, 2012; Sadee, 2022; Sathiyamoorthy et al., 1997; Tunçtürk et al., 2015; Turan et al., 2003). These studies have mostly focused on areas representing a single source of pollution (mining sites and non-residential or non-agricultural areas) and have concentrated on elements such as Cd, Cu, Fe, Pb, Ni, and Zn.

Studies examining heavy metal levels in soils and WEPs in areas with altered land use are quite limited. On the other hand, it has been reported that land use has undergone significant changes globally over time. Indeed, Winkler et al. (2021) estimated that changes in land use during the period 1960–2019 affected approximately 32% of the global land area. Furthermore, between 2000 and 2019, the expansion of industrial land in developing countries contributed 31% to economic growth and 55% to emissions (Yoo et al., 2024).

When land use changes, meaning when urban, agricultural, and industrial areas become intertwined or replace one another, how the heavy metal content of WEPs collected from soil and nature is affected by these changes is an important research topic, particularly in terms of human health, and the findings will shed light on future spatial planning efforts.

Therefore, this study aims to determine the extent of heavy metal accumulation in GT plants and soils grown in the Kemalpaşa region, which has historically been used for agricultural production and currently hosts 17 different industrial production sectors across an area of 130 km2, identify the possible sources of these metals, and assess the potential human health risks associated with the consumption of GT through the Mediterranean diet.

With its broad scope, the study contributes to in-depth discussions on how the environmental impacts of industrial zones and changes in land use affect ecology and human health. Thus, it provides evidence that can inform spatial planning, agricultural management, and food security policies in industrialized regions.

Materials & Methods

Study area

Soil and plant sampling was carried out near Kemalpaşa County, Izmir Province. The study area is located in the west of the Anatolian Peninsula with a total area of 658 km2 and a population of 117,956 people (Kemalpaşa Municipality, 2024; TurkStat, 2024). The region has a typical Mediterranean climate; summers are hot and dry, winters are mild and rainy. The dominant wind direction is south and southwest throughout the year. The average annual rainfall is 1,050 mm and snowfall is almost non-existent (Gül, 2005). Approximately 60% of the plain (336.11 km2) is covered with forests, while 229 km2 is agricultural land (Kemalpasa District Governorship, 2016). To investigate the effects of heavy metals originating from complex anthropogenic activities on human health through edible plants, it was decided to study an area that (i) has an agricultural history, (ii) is intensively operated by different industrial sectors, and (iii) is a region where wild edible plant species grow widely. The Kemalpaşa region has an agricultural history, is densely populated by industrial facilities from different sectors, and the edible wild plant GT was also commonly observed in this region. In this respect, the sampling site is suitable for the research hypothesis.

It is known that the first village settlements in Kemalpaşa date back to the Late Neolithic period in Western Anatolia (6500–5000 BC), that traditional agriculture continued until the 1930s, and that the first artificial fertilizers and agricultural chemicals began to be used following the establishment of the Bornova-Izmir Plant Protection Research Institute in 1931 and the Turkish Agricultural Control and Quarantine Organization in 1957 (Evni & Kan, 2024; Izmir Provincial Culture and Tourism Directorate , 2025; Kadıoğlu, 2012). Established in 1992 on the agricultural areas in the western part of the plain, the total area of İzmir Kemalpaşa Organised Industrial Zone (KOSBI) has reached 130 km2 as of 2012. For this reason, it offers an important intertwined economic structure with industry in the west and agricultural areas in the east (Kemalpaşa Municipality, 2024).

In addition to industrial activities from 17 different sectors, there are fuel stations and storage areas in KOSBI (2023b). Industrial establishments generally supply the water they use in their production activities from underground wells in their own sites. The central wastewater treatment plant in the region became operational approximately 18 years after the start of industrial activities, and the wastewater of the majority of the establishments is collected by a central system and treated in this facility. However, the enterprises that are not connected to this system discharge the wastewater they process in their own treatment plants to the surrounding surface waters (KOSBI, 2023b). The annual import and export capacity of the region is approximately 2 and 3 billion USD, respectively. There are 421 enterprises in KOSBI and 82% of them are industrial enterprises. Of the industrial enterprises, 24% operate in the machinery and 15% in the iron and steel processing sectors. Other prominent sectors are building materials (11%), chemistry (11%), automotive supply industry (10%), plastics (10%) and food (5%) (KOSBI, 2023a). The locations of the main industrial facilities with high atmospheric emissions and water use are presented in Fig. 1 (Figs. S1–S9).

Figure 1 Sectoral distribution of industries in KOSBI and sampling locations (Map data: Google, ©2024, Maxar Technologies).

Sampling

Plant and soil sampling were carried out in the spring of 2023. No environmental matrix other than soil and plants was included in the study, which contributed to reducing confounding factors. In this study, GT, which grows widely in the region, is collected from nature by the people and consumed frequently, was selected as a sample plant to obtain ecological validity (Hani et al., 2024; Konak, Ateş & Şahan, 2017; Tarhan, Firat & Topal, 2023). Although GT is a perennial plant, it regenerates every year, and its roots and stems mature and are consumed between March and May (Danin & Fragman-Sapir, 2007). Sampling points (n = 13) were determined by taking into consideration the areas with research permission and suitable for both soil and plant sampling. To minimize sampling bias, representative areas were carefully selected within the study area. Samples were not taken from inside industrial facilities or roadside areas in order to reduce the impact of local pollution sources. Instead, unused plots that were less affected by direct human activity but still suitable for reflecting the actual conditions in the region were preferred. In order to obtain background (reference) values for the region, reference soil and plant samples were taken from a point away from industrial influence, downwind of the prevailing wind directions and at least 300 m higher than the sampling site (Fig. 1).

Soil samples were shoveled from the first 10 cm of the surface and placed in paper bags. Plants were removed from the soil together with their roots and placed in separate paper bags. All samples were transported to Dokuz Eylü l University Solid Waste and Soil Pollution Laboratory on the same day. All soil and plant samples were collected and stored following the same procedures.

Sample preparation

Soil samples were dried at 40 °C for 24 h and then sieved through a two mm sieve to remove the coarse fraction. The homogenized fine fraction was used in the analysis. Soil adhering to the root parts of GT plants was removed using a plastic brush. After this step, the plants were washed under running tap water and rinsed with distilled water, then the edible parts of the root and stem were separated and each was divided into two equal parts. One part was air dried at room temperature for 5 days for elemental analyses and the other part was used for moisture determination in order to calculate metal concentrations according to fresh weight. The dried plant samples were ground to <2 mm particle size using an agate hand mortar.

Analytical methods

Moisture and organic matter contents of soil and plant samples were determined in accordance with TS ISO 11465 (Turkish Standards Institute, 2015) and ASTM D2974-13 (ASTM International, 2014) standards, respectively. Soil pH and electrical conductivity (EC) values were measured with WTW Multi 3620 IDS device in the supernatant of the suspension prepared with 1/5 solid/liquid ratio and shaken for 5 min.

Cd, Cr, Ni, Pb and Ti elements below 10 ppm in soil and plant samples were analyzed by ICP-MS (Agilent 7850 ICP) and Cu, Zn, aluminum (Al), Fe, K, Na, magnesium (Mg), Mn, P and S elements were analyzed by X-ray Fluorescence Spectrometry (XRF, Spectro IQ II). The detection limits of both devices for the relevant elements are presented in Table S5.

Pollution assessment in soils and plants

Enrichment factor

The enrichment factor (EF) indicates the extent to which a given element has accumulated in the soil compared to the background value. The EF value was calculated using the formula in Eq. (1) (Buat-Menard & Chesselet, 1979). (1) EF=CE−S/CRE−S/CE−AEC/CRE−AEC

where: CE−S is the concentration of the studied element in sample; CRE−S is the concentration of the reference element in the sample; CE−AEC is the concentration of the studied element in reference material; CRE−AEC is the concentration of the reference element in reference material. Reference soil sample was used as the reference material, and the crustal element of Mg was used as the reference element since its level in reference soil is below its abundance value in the Earth’s crust (AEC) (Champion, 2008).

The EF value is used to classify the degree of enrichment (Akinci, Gok & Bilgin, 2019; Loska, Wiechuła & Pelczar, 2005; Sutherland, 2000; Yaylali-Abanuz, 2011)

EF <2: Deficiency or minimal enrichment

2 ≤ EF <5: Moderate enrichment

5 ≤ EF <20: Significant enrichment

20 ≤ EF <40: Very high enrichment

EF >40: Excessive enrichment

Pollution index

Pollution index (PI) was calculated to estimate the level of multi-element pollution in soils (Tarcan, Akıncı& Danışman, 2010; Zhang et al., 2021). PI values for Cd, Cr, Cu, Ni, Ni, Pb, Zn and S were calculated according to Eq. (2). (2) PI=Cd/3+Cr/100+Cu/100+Ni/50+Pb/100+Zn/300+S/300/7.

Tolerable level values were taken from the literature values reported by Gemici (2008) and Sponza & Karaoǧlu (2002). PI > 1.0 indicates the presence of anthropogenic pollution in the soil.

Transfer of elements from soil to plant parts

In order to evaluate soil-plant heavy metal transfer, bioconcentration factors (BCFr and BCFs) explaining the transport to the root and stem were calculated according to Eqs. (3) and (4), respectively. (3) Bio-concentrationfactorforrootBCFr=Metalinrootsmgkg/Metalinsoilmgkg

(4) Bio-concentrationfactorforshootBCFs=Metalinshootmgkg/Metalinsoilmgkg.

If the BCF value is above 1, the plant is considered to be a hyperaccumulator of the relevant metal (Ahmad et al., 2023; Akinci, Gok & Bilgin, 2019; Lai et al., 2023; Lai et al., 2024; Liu et al., 2015).

The ability of the plant to transport metal from the root to the upper tissues was evaluated by translocation factor (TF) (Ahmad et al., 2023; Malik, Husain & Nazir, 2010; Moameri et al., 2017). When TF < 1, the plant is stressed and cannot efficiently transport the metal to the aerial parts of the plant; TF > 1 indicates that the plant can transport this metal to the upper tissues and is probably a hyperaccumulator (Akinci, Gok & Bilgin, 2019; Brunetti et al., 2009). TF value was calculated with Eq. (5). (5) TranslocationfactorTF=Metalinshootmgkg/Metalinrootsmgkg.

Statistical analysis

Due to the relative (compositional) nature of the elemental concentration data, a centered log-ratio (clr) transformation was applied prior to correlation analyses. This transformation was performed by calculating the geometric mean of the elemental concentrations in each sample and then taking the natural logarithm of the ratio of each element concentration to this mean. Thus, distortions due to the closure problem common in compositional data were avoided and valid application of Pearson correlation coefficients was possible (Khan et al., 2024). All analyses were performed on clr-transformed data sets using SPSS version 25.0 (IBM Inc., Armonk, NY, USA) software.

Correlation analyses were carried out to identify the interrelations among the levels of all studied elements in soil, plant root and plant stem. A principal component analysis (PCA) was also applied to identify the possible sources of observed elemental levels. Input variables are; Al, Cd, Cu, Fe, K, Mn, Ni, P, Pb, S and Zn for soil, Cd, Cr, Cu, K, Na, Ni, P, Pb, S and Zn for root and Al, Cd, Cr, Cu, K, Mn, Na, P, Pb, S and Zn for stems. PCA was applied to the data sets as the extraction method, and the raw calculated factor loading coefficients were rotated by Varimax with a Kaiser Normalization. Kaiser–Meyer–Olkin Measure of Sampling Adequacy and the significance of Bartlett’s Test of Sphericity of all data sets were calculated as >0.500 and <0.001, respectively; therefore, the sample sizes were found suitable to be evaluated using PCA for soil, for plant root and the stem. In addition, the principal components (PCs) with clusters defined by factor loading coefficients ≥ 0.5 and with Eigenvalues greater than 1.0 were extracted.

The PCA results obtained for soil, root, and stem samples were evaluated using the principal component scores for each sampling point. The factor scores indicate the positions of the samples on the principal component axes, and these scores were used to determine the relationships between the sampling points and the element groups, as well as their degree of separation from each other. The scores were standardized as z-scores, with —score—≥ 1.0 classified as “significant,” 0.5−1.0 as “moderate,” and 0−0.5 as “weak” effect levels. This approach allowed for the comparison of different sampling areas based on element accumulation patterns and the identification of spatial distributions.

Human health risk assessment

Nemerow composite pollution index

Nemerow Composite Pollution Index (NCPI) is the mathematical expression commonly used to calculate the integrated contaminant index when contaminants are present in food samples (Proshad & Idris, 2023; Román-Ochoa et al., 2021). For NCPI, the single factor index (Pi) and the average Pi value were calculated. Finally, the composite index was assessed using the average and maximum index values (Pi−mean and Pi−max) (Proshad & Idris, 2023; Wei & Cen, 2020). Eq. (6) was used to calculate Pi. (6) Pi=CiSi.

In Eq. (6), Pi represents the pollution index of a single element; Ci indicates the metal content in food sample, and Si represents the maximum permissible individual metal concentration for a certain food group (Table 1). In addition, if the Pi value is ≤1, the food sample is safe, while if the Pi value is >1, the food sample is evaluated as contaminated.

Table 1 Maximum permissible concentration (MPC) of metals in vegetables (mg/kgfw*).

Heavy metals	Cd	Pb	Cr	Zn	Fe	Cu	Ni	Mn	
Permissible Concentration	0.1a,b	0.1a,b	2.3c	20d	45d	10d	1.5d	16.61d	
Notes.

a (The Ministry of Agriculture and Forestry, 2023).

b (Food and Agriculture Organization of the United Nations, 2023b).

c (Shaheen et al., 2016).

d (Kharazi et al., 2021).

* fw, (fresh weight).

Nemerow Composite Pollution Index can be calculated by Eq. (7). (7) NCPI=Pi−mean2+Pi−max22.

Pi−mean represents the average value of the single factor index and Pi−max represents the maximum value of the single element contamination index. Food is classified according to NCPI values as follows: NCPI<1.0: uncontaminated food, 1.0≤NCPI<2.5: lightly contaminated food, 2.5≤NCPI<7: moderately contaminated food and NCPI≥7: heavily contaminated food (Proshad & Idris, 2023; Román-Ochoa et al., 2021).

Estimated daily intake of heavy metals

Daily heavy metal intake of individuals (EDI) was assessed according to nutritional needs. The daily intake of heavy metals from vegetables can be assessed based on metal contents, food ingestion rate, and the weight of people. Equation (8) is used to calculate EDI (Kukusamude et al., 2021; Román-Ochoa et al., 2021). Then, EDI values were compared with tolerable upper intake levels (TUILs) obtained from US Food and Nutrition Board Institute of Medicine and FAO/World Health Organization (WHO) (Food and Agriculture Organization of the United Nations, 2023b; U.S. Food and Nutrition Board Institute of Medicine, 2002). (8) EDI=Ci×EF×ED×DCBW×TA×10−3

where Ci (mg/kgfw) is the concentration of the heavy metal in a food sample; EFrepresents the exposure frequency (times per year); ED is the exposure duration (5 years for children and 41 years for adults) (Kukusamude et al., 2021); DC represents the average daily GT consumption (grams per day). BW is the body weight which is expressed as 73.7 kg for an adult (TurkStat, 2023) and 16 kg for a child (Proshad & Idris, 2023) and TA is the average exposure time for non-carcinogenic (ED x 365 days/year) (Kukusamude et al., 2021).

Although the frequency of wild greens consumption in Southern Europe is reported in the literature as twice a week (Leonti et al., 2006), GT is consumed between March and May (Danin & Fragman-Sapir, 2007; Hani et al., 2024). For this reason, the EF value was accepted as 24 times per year in this study. Assuming that a meal made from one kg of untrimmed GT plant serves four people in local cuisine recipes, the DC value has been used as 200 g per day (Nefis, 2014).

Noncarcinogenic health risk assessment

In order to assess the potential effects of heavy metals detected in plants on human health, non-carcinogenic risk assessment was conducted in accordance with United States Environmental Protection Agency (USEPA) standards (USEPA, 2002). Non-carcinogenic risk was assessed by calculating the Estimated Daily Intake (EDI) for each metal and then divided by the respective reference dose (RfD) to obtain the Hazard Quotient (HQ) (Eq. (9)) (Hasan et al., 2021; Khan et al., 2022; Li et al., 2014). To estimate the combined risk, the HQs for all metals were summed to obtain a Hazard Index (HI) (Eq. (10)). An HQ or HI greater than 1 indicates the potential for non-cancer adverse effects (Hasan et al., 2021; Khan et al., 2022; Li et al., 2014). (9) HQ=EDIRfD

(10) HI= ∑HQi.

Oral RfD values for Cd, Cr, Mn, Ni and Zn were obtained from USEPA and were 0.001, 0.0009, 0.14, 0.02 and 0.3 mg/kg day, respectively (USEPA, 2024b). In addition, RfD values of 0.04 mg/kg day for Cu (USEPA, 2024c), 0.7 mg/kg day for Fe (USEPA, 2006), and 0.0014 mg/kg day for Pb (Hasan et al., 2021; Li et al., 2014) were obtained from the relevant literature.

Carcinogenic risk assessment

Carcinogenic risk assessment assesses the lifetime probability of developing cancer due to exposure to carcinogens. For metals classified as carcinogenic (International Agency for Researchon Cancer, IARC 2025), the lifetime cancer risk (CR) was calculated by multiplying the EDI by the cancer slope factor (SF) of the metal (Eq. (11)) (Hasan et al., 2021; Khan et al., 2022; Li et al., 2014). SF values were obtained as 15 mg/kg day for Cd (California Office of Environmental Health Hazard Assessment OEHHA, 2009), 0.27 mg/kg day for Cr (USEPA, 2024a), 0.09 mg/kg day for Ni (OEHHA, 2009) and 0.0085 mg/kg day for Pb (Hasan et al., 2021). Acceptable CR values are generally in the range 10−6 to 10−4. These calculations provide a quantitative basis for assessing potential health hazards from long-term exposure to contaminated soil and plant materials (Hasan et al., 2021; Khan et al., 2022; Li et al., 2014). Total risk of multiple metals was calculated with Eq. (12) (Hasan et al., 2021). (11) CR=EDI×SF

(12) CRTotal= ∑CRi.

Results and Discussion

Properties of soil samples

The pH, electrical conductivity and organic matter contents of the soil samples are given in Table S6. It was observed that the soils had a pH range between slightly acidic and neutral, and all were in the non-saline soil group (Canada Government, 1998; Chhabra, 2005; Loganathan, 1987; Neina, 2019; United States Department of Agriculture, 1999). The organic matter content of the soils is generally at a moderate level, with very high levels of organic matter in sample S1 (Montgomery et al., 2022; Muminov et al., 2018).

Soil pollution assessment

Heavy metals and other elemental levels in soils

The concentrations of heavy metals and other elements in samples S1–S13 and the reference soil (REF-S), along with their AECs, are provided in Table S3 (Champion, 2008). The concentrations of Cu, Ni, Zn, Al, Fe, and Ti in the soil samples were 806–1,463 mg/kgdw, 105.11–188.22 mg/kgdw, 77.90–2,421 mg/kgdw, 92,600–123, 800 mg/kgdw, 64,840–96,380 mg/kgdw, and 6,050.51–8,955.75 mg/kgdw, respectively, and that these values were above both REF-S and AEC values. In all samples, Mn (1,081–1,963 mg/kgdw) and P (1,689–3,675 mg/kgdw) concentrations were above the AEC values and exceeded the reference levels in all samples except S11. Cd levels (0.15–35.80 mg/kgdw) were found to be equal to or higher than the AEC; additionally, Cd concentrations in samples S1, S3, S4, S6, S10, and S11 were above the REF-S value. In terms of Cr, concentrations in samples S3, S5, S10, and S13 were above both REF-S and AEC values, while in samples S1, S2, S4, S8, and S9, they were only above the REF-S value. Pb concentrations exceeded both REF-S and AEC values in samples S1, S2, S3, S6, and S12. K levels (16,880–26,460 mg/kgdw) were measured above the values specified in REF-S and AEC in many samples. Na and Mg concentrations were generally below the AEC. However, Na in S1 and Mg in S1, S2, S3, and S8 exceeded the REF-S value. Furthermore, in nine samples (S1, S2, S3, S4, S5, S7, S10, S12, and S13), soil S concentrations were higher than REF-S, and among these samples, only S1 (1,163 mg/kgdw) exceeded the AEC value. In a general assessment, it was determined that levels of eight or more elements in all 13 soil samples were above the levels observed in REF-S, clearly demonstrating the presence of elemental contamination in the region’s soils.

Previous studies on the element levels of soil samples taken from GT-grown areas have been limited to arid regions and Pb, Cu, and Pb-Zn mines. Soil Cu values determined in mining areas ranged from 10 to 450 mg/kgdw, while Ni, Pb, Zn, Cd, and Fe metals ranged from 6 to 1,730 mg/kgdw, 22 to 16,700 mg/kgdw, 8–2,950 mg/kgdw, 0.2–81 mg/kgdw, and 80–1,730 mg/kgdw, respectively (Chehregani, Mohsenzade & Vaezi, 2009; Ghaderian & Ghotbi Ravandi, 2012; Moameri et al., 2017; Mohsenzadeh & Rad, 2012). Compared to studies conducted on soils around mining sites, this study found much higher levels of Cu (806–1,463 mg/kgdw) and Fe (64,840–96,380 mg/kgdw) and much lower levels of Pb (1.01–1,157.64 mg/kgdw), while Zn (77.90–2,421 mg/kgdw) and Ni levels (105.11–188.22 mg/kgdw) are close to the reported values (Chehregani, Mohsenzade & Vaezi, 2009; Ghaderian & Ghotbi Ravandi, 2012; Moameri et al., 2017; Mohsenzadeh & Rad, 2012). The lowest Ni value observed in this study is above the lowest levels encountered in mining areas, but the maximum value is lower than the levels measured in soils around mining areas. The Cd, Cu, Ni, Pb, and Zn levels in soil samples taken from an arid region where GT grows are much lower than the levels detected in this study (Sathiyamoorthy et al., 1997).

Soil pollution factors

Enrichment factor (EF) is used to assess the elemental pollution of the soils from study area with respect to their concentrations in REF-S according to Eq. (1) (Table 2). EF values of Cu, Al, Fe, Na, Mn and P of samples express a profile in the scale of deficiency to minimal enrichment (0.06≤EFCu,Al,Fe,Na,Mn,P ≤1.89) in soils. Ni shows significant enrichment in all soil samples (6.93≤EFNi ≤11.51) and may originate from the iron and steel processing, chemical production, machinery manufacturing, electrical-electronics, and automotive industries, which constitute more than 72% of the industrial establishments in the study area. Of the 13 soil samples taken, 10 have an EF >  2 value for Zn; in three samples (S5, S7, S12), EFsZn is in the range of 3.39−4.2, indicating moderate enrichment, while in six samples (S2, S3, S4, S6, S11, S13) range from 5.75 to 13.06, indicating significant enrichment, while S1 was calculated as 30.93, indicating very high enrichment. The main industrial sources of Zn are iron and steel processing, plastics processing, machinery manufacturing, electrical-electronics, and automotive industries. These industries are also very common in KOSBI and represent more than 60% of active industries. Cr is another heavy metal whose EFs were frequently determined to be >2 in soil samples; EFsCr was found to be between 3.32 and 4.18 (moderate enrichment) in S1, S2 and S8, and between 6.03 and 10.37 (significant enrichment) in S3, S4, S5, S9, S10 and S13. Iron and steel, machinery manufacturing, automotive, chemical and electrical-electronic sectors, which constitute more than 61% of the total share of the sectors operating in the KOSBI region, can be considered as possible industrial origin Cr sources. With EFPb of 81.34 in S1 soil, Pb was the only heavy metal expresses extremely high enrichment among others. Moreover, EFPb in S3 (EFPb = 31.75) indicated very high enrichment. Pb enrichments in S2 (EFPb = 16.53), S6 (EFPb = 8.59) and S12 (EFPb = 12.88) were in the category of significant enrichment. S1, S2, S3 and S12 are the sampling points which are the closest to the paper industries in the region (Fig. S1).

Table 2 Elemental enrichment factors (EFs) determined for soil samples.

	Soil Samples	
Elements	S1	S2	S3	S4	S5	S6	S7	S8	S9	S10	S11	S12	S13	
Cd	1.04	0.05	1.56	1.85	0.02	2.86a	0.01	0.01	0.01	0.99	1.20	0.02	0.02	
Cr	3.77a	3.32a	9.17b	6.94b	10.37b	0.20	0.19	4.18a	6.03b	6.84b	0.15	0.16	9.12b	
Cu	1.59	1.25	1.42	1.17	1.28	1.57	1.45	1.61	1.81	1.79	1.89	1.51	1.21	
Ni	9.00b	8.57b	10.99b	9.00b	11.51b	9.37b	8.93b	8.86b	9.43b	8.96b	8.31b	6.93b	10.36b	
Pb	81.34d	16.53b	31.75c	0.18	0.10	8.59b	0.10	0.09	0.16	0.16	0.14	12.88b	0.12	
Zn	30.93c	8.43b	6.84b	13.06b	4.20a	7.82b	3.39a	1.85	1.30	1.90	5.75b	3.95a	6.69b	
Al	1.05	1.20	1.43	1.57	1.74	1.37	1.20	1.22	1.27	1.17	1.32	1.25	1.55	
Fe	1.39	1.16	1.56	1.36	1.77	1.36	1.28	1.47	1.24	1.26	1.28	1.21	1.49	
K	1.84	1.88	2.09a	2.41a	2.57a	1.96	1.76	1.77	1.58	1.77	1.64	1.81	2.19a	
Na	1.07	0.06	0.06	0.07	0.08	0.19	0.12	0.49	0.07	0.06	1.10	0.71	0.08	
Mn	1.12	1.04	1.63	1.36	1.59	1.35	1.07	1.29	1.20	1.29	1.07	1.32	1.40	
P	1.89	1.14	1.13	1.59	1.57	1.23	1.20	1.01	1.25	1.29	1.14	1.28	1.48	
S	5.01b	0.92	1.32	1.85	1.69	1.05	1.47	0.74	0.92	1.02	1.04	1.67	1.37	
Ti	1.71	2.08a	2.13a	2.22a	2.93a	1.75	2.13a	1.82	2.30a	2.29a	1.93	2.16a	2.18a	
Notes.

Scale: EF<2, Deficiency to minimal enrichment.

a 2<EF<5: Moderate enrichment.

b 5<EF<20: Significant enrichment.

c 20<EF<40: Very high enrichment.

d EF>40 Extremely high enrichment.

S enrichment in S1 was in the category of significant enrichment, while S EFs in other samples were <2. It is known that S pollution mainly originated from coal and petroleum combustion (Bradl, 2005) which is a very common energy source for the processes of the industries in KOSBI. However, it should be noted that S1 is the closest sampling point to the Kemalpaşa, where natural gas distribution system is recently obtained to the town and most households prefer coal for heating purpose (Natural Gas, 2022). The highest EFCd (2.86, moderate enrichment) was calculated in S6, where high number of machinery manufacturing and iron and steel industries are located in its vicinity. EF levels for K, which is a crustal element, in S3, S4, S5 and S13 show moderate enrichments; there are industries operating in construction and materials production close to these sampling locations. EFs calculated for Ti in many soil samples showed moderate enrichments (2.08≤EFTi ≤2.93); the sampling points with higher EFTi values were calculated for the samples close to one or more industries producing chemicals (dyes) and/or construction materials.

The EF values calculated for Cd and Cu in soil samples are significantly higher than those determined in studies conducted at thermal power plants, industrial cities, and organized industrial zones. The EF values determined for Cr and Mn in the soil in this study were below those determined at a thermal power plant and in the soil of another organized industrial zone in Türkiye, and were found to be close to those obtained in the soil of an industrial port city (Akinci, Gok & Bilgin, 2019; Almasoud, Usman & Al-Farraj, 2015; Yaylali-Abanuz, 2011). EF values for Ni are significantly higher than those obtained in previous studies in regions with intensive industry, EFPb values are within the range of values determined in these regions except S1, and EFK values are in a similar range, while EFS values are much lower.

PI was used to assess the extent of pollution from multiple elements and varied between 2.06 and 6.82 (Fig. 2, Table S7), which explained that the studied soils can be contaminated by anthropogenic sources. The ranking of sampled soils according to the PI values was as follows: S1 > S3 > S6 > S10 > S4 > S11 > S2 > S8 > S9 > S12 > S7 > S13 >  S5. Neither the number of different types of industries nor the total number of industries in the sampling vicinity were found correlated with calculated PI values. In a study conducted in Aliağa, an industrial zone where iron and steel production is intense, soil PI values were found between 0.32 and 3.5 (Sponza & Karaoǧlu, 2002), where PI values were found between 0.6 and 31.5 in upstream and downstream river bottom sediments in an abandoned mercury mine (Gemici, 2008). In another study was conducted on soils in an area affected by tannery tailings, PI values were found between 0.38 and 5.37 (Tarcan, Akıncı& Danışman, 2010). In general, the pollutant sources investigated in the compared literature are more prominent and soil PI values are lower than those in this study.

Figure 2 Elemental pollution index values (PI) detected in soil samples (Map data: Google, ©2024, Maxar Technologies).

Heavy metals and other elemental concentrations in plants

Heavy metals, Al, Fe, K, Na, Mg, P, S and Ti were measured in root and stem samples of GT and the results are given in Tables S8 and S9, respectively. Cd concentrations were found to be slightly higher than the reference root sample (REF-RO; 0.04 mg/kgdw) in samples RO6, RO8, RO9, RO10, RO11, RO12, and RO13. Cr was below the reference (1.11 mg/kgdw) only in RO1, RO2, RO3, and RO4, while Ni was above the reference (1.43 mg/kgdw) in RO8, RO9, RO10, and RO12. Pb was below the REF-RO value (2.46 mg/kgdw) in all samples. In all root samples, Zn and Na concentrations were found to be above the reference, while K was below the reference. Ti was found to be below the reference values only in RO1, RO4, RO6, and RO7, and Cu was found to be below the reference values in RO5, RO11, and RO13. Al (RO3, RO6), Mn (RO2, RO9, RO10), and Fe (RO1, RO2, RO5, RO9, RO13) were found to be higher than the reference level in some samples. Mg was found to be lower than the reference level only in RO6, while P (RO13) and S (RO3, RO5, RO6, RO9, RO12) were found to be below the reference level in some samples. Overall, Zn was the heavy metal most frequently found to exceed the reference level, while Al (in 11 samples) and Cu (in 10 samples) were also frequently found in high concentrations. Sample RO9 exceeded the reference level for Cd, Cr, Cu, Ni, Zn, Al, Fe, Mn, and Ti, while each of the other root samples exceeded the reference level for at least three heavy metals. In all GT stem samples, Cu and Ti levels were higher compared to the reference sample (REF-ST; Cu: 1410 mg/kgdw, Ti: 0.28 mg/kgdw). Similarly, Cr and Pb concentrations exceeded the reference values in all samples except ST10. Cd levels were slightly higher than REF-ST (0.17 mg/kgdw) in seven samples (ST1, ST2, ST3, ST6, ST7, ST9, ST12), while Ni was below the reference value only in ST6, ST10, and ST12 and above the reference value in other samples. Zn was detected in ST1, ST2, and ST6; Fe was detected in all samples except ST6 and ST12; Al, Na, and S were detected above the reference value in all samples except ST6. Mn was found to be lower than REF-ST in all stem samples, while Mg was found to be low only in ST3, ST6 and ST10 and high in the others. K concentrations were low in five samples (ST1, ST3, ST11, ST12, and ST13) and high in the others, while P exceeded the reference value in ST4, ST5, ST10, and ST12. Consequently, samples ST1 and ST2 showed above-reference values for Cd, Cr, Cu, Ni, Pb, Zn, Al, Fe, and Ti, while the other samples exhibited enrichment in at least four heavy metals.

Heavy metal (Cd, Cr, Cu, Ni, Pb, Zn, Fe, Mn) levels measured on fresh weight in GT roots and stems were compared with the maximum permissible concentrations (MPCs) determined for vegetables (Tables S10 and S11). In roots (including REF-RO), Cu, Pb, Zn, Fe, and Mn were found above the MPC, while Cd, Cr, and Ni were found below the MPC. In all stem samples (including REF-ST), Cu (107.87–244.97 mg/kgfw), Fe (47.84–293.61 mg/kgfw) and Mn (25.38–52.53 mg/kgfw) values were found above the MPC values reported for vegetables (Cu: 10 mg/kgfw, Fe: 45 mg/kgfw, Mn: 16.61 mg/kgfw). In contrast, Cd (0.01–0.06 mg/kgfw), Cr (0.07–0.23 mg/kgfw), and Ni (0.05–0.24 mg/kgfw) levels in the stems were below the MPC. Pb exceeded the MPC (0.1 mg/kgfw) in samples ST1, ST2, ST3, ST4, ST5, and ST11. The MPC for Zn (20 mg/kgfw) was not exceeded only in samples ST7, ST8, ST9, and ST13.

In the literature, Cu levels of 11–24 mg/kgdw, Pb levels of 4–652 mg/kgdw, Zn levels of 13–820 mg/kgdw, and Fe levels of 1952 mg/kgdw have been reported in GT plants grown at mining sites. Cd and Ni concentrations ranged from 0.48–2.30 mg/kgdw and 0.6–23.33 mg/kgdw, respectively (Chehregani, Mohsenzade & Vaezi, 2009; Ghaderian & Ghotbi Ravandi, 2012; Moameri et al., 2017; Mohsenzadeh & Rad, 2012). In this study, Cu (1,410–6,330 mg/kgdw), Zn (155–3,053 mg/kgdw), and Fe (82–6,480 mg/kgdw) levels in GT samples were significantly higher than those reported at mining sites. On the other hand, Cd, Ni and Pb levels were found to be lower compared to GT samples from mining sites.

Elemental bioconcentration and translocation in plants

The transfer of heavy metals from the soil to the plant roots (BCFr) and stems (BCFs) was evaluated by calculating the bio-concentration factors (Table S12). BCFr and BCFs values for Cu, K, Na, Mg, P and S were found to be >1.0 in all samples. BCFr values for Zn were calculated as >1.0 in all samples, while BCFsvalues calculated for Zn were >1.0 for ST2, ST3, ST5, ST6, ST8, ST9, ST10, ST11, and ST12. BCFr values calculated for Mn were >1.0 for only RO2 and RO9. Moreover, BCFsvalues for Cd in ST9 and ST11 and BCFsvalue for Cr in ST11 were >1.0. BCFr values calculated for Pb were >1.0 for RO5, RO7, RO8, RO9, RO11 and RO13. BCFsvalues for Pb were >1.0 for only ST5 and ST11.

When the standard deviations (%) of BCFs calculated for each element in the samples were analyzed, it was observed that Cd, Cr, Ni, Pb, Zn, Fe, Na, S and Ti were distributed in a wide range around the mean value (40.84%–185.79%), while the % standard deviations were lower for other elements (Table S12). High standard deviations mean that these elements may have many sources in the study area and/or the geochemical features of soils show heterogeneity (Akinci & Ozuysal, 2022). For example, as discussed in soil pollution factors section, Fe and Na are among the elements having soil EFs <2.0, which indicates deficiency to minimal enrichment, therefore, the high standard deviations calculated for the BCF values of these elements may be partially associated with the geochemical heterogeneity in the region. BCF values >1 in GT samples are rarely observed in studies carried out at mining sites, as there is only one main source of the elements (Table S4). In this study, BCFs >1 for Cd, Cr, Cu, Pb, Zn, K, Na, Mg, Mn, P and S were observed in at least one sample and BCF values as high as 206 (for Na at ST13) were also encountered in the study area where many industrial plants are actively operated. Therefore, elements such as Na and S, which generally have low EFs in soils but have high BCFs in the aerial parts in some samples, may indicate the influence of strong anthropogenic sources in the region.

BCF values determined for Cu in GT samples grown in an arid region without the presence of anthropogenic sources are similar to the values obtained in this study, while GT-BCF values obtained in studies conducted around Pb and Cu mines are lower (Chehregani, Mohsenzade & Vaezi, 2009; Ghaderian & Ghotbi Ravandi, 2012; Mohsenzadeh & Rad, 2012; Sathiyamoorthy et al., 1997). The same researchers found the BCF value for Zn to be very high in the arid area and low in mining areas compared to this study, while BCF values calculated for Fe were higher in the vicinity of Pb mines. BCF values for Pb calculated in this study were below the study conducted in an arid region. In this study, the calculated BCF values for Ni were generally lower than those reported in studies conducted at mining sites and in the arid zone. Moreover, the BCF values for Cd were very high in the arid area and low in the mining areas compared to this study.

The capacity of the GT to transport metals from the roots to the aerial parts was assessed using the translocation factor (TF). TF values of Cu, Zn, K, Mn and P in sampling points (S1-S13) were <1.0, indicating poor transfer of the corresponding element to the aerial parts due to low toleration or metal stress. All TF values for Cd were >1.0 in all sampling points, where TFs for Cr were also calculated >1 in S1, S2, S3, S4, S5, S11, and S13. In sampling points S4, S7, S8, S11, S12 and S13, TFs between 1.0−2.26 were calculated for Al, Na, Mg and S, while TFs for Fe were >1 in S3, S4, S10 and S11.

Moameri et al. (2017) has investigated the bioconcentration capacity of Cd, Ni, Pb and Zn with their translocation factors in Pb-Zn mining site. It was concluded that GT is a low accumulator for Cd (TF: 0.32), Ni (TF: 0.89), Pb (TF: 0.16) and Zn (TF: 0.09), while the average TF values for Cd (21.76), Ni (3.99), Pb (0.77) and Zn (0.51) here was found to be much higher (Table S12). The main factors influencing these very different TF values calculated for the same plant in different study sites are the elemental content in the surrounding soil and air, bioavailability, pH, cation exchange capacity, climatic conditions, and growing season (Petrović, Medunić & Fiket, 2023)

Statistical evaluation of elemental data

Inter-correlations between the elemental levels detected in soils and plant parts

Pearson correlation coefficients were calculated between the element levels detected in the studied soil and plant matrices as a first step and secondly, the inter-matrix relationships of the elements were discussed.

Al, Cu, Fe, K, Mg, Mn, Ni, P and Ti levels in soils are well correlated with each other (p ≤ 0.05), while Cd, Pb and Zn levels are negatively correlated with the elements in this group, but not with each other (Table S13).

In GT roots, Ni was found to have statistically significant strong correlations with Cd and Cr (p ≤ 0.01), but Cd and Cr did not correlate with each other (Table S14). On the other hand, these three heavy metals showed strong negative correlations with Na and Mg (p ≤ 0.05). Cu in GT roots showed strong correlations with K, Mn, P and Zn (p ≤ 0.05). Sulphur in GT roots had the only positive correlation with Zn (p ≤ 0.05) and a negative correlation with Ni (p ≤ 0.05). Al, Na and Mg in the root parts of GT from the sampling points were also correlated with each other (p ≤ 0.05).

Similar with roots, Cu in GT stems showed statistically significant strong correlations with K, Mn and P (p ≤ 0.01) and Al with Na and Mg (p ≤ 0.01) (Table S15). It was seen that Cr has negatively correlated with Cu, K, Mn and P. Stem Pb levels showed strong correlations with Cd, Cr and Zn (p ≤ 0.05), which have not correlated with each other. Zn also had negative correlations with Al, Fe, Na and Ti (p ≤ 0.05) in GT stems.

Soil and GT root Zn levels were well correlated (p ≤ 0.05), but this was the only one-to-one correlation that could be observed between the concentrations of the same element in soil and GT root zone (Table S16). In addition to that, root Zn levels were negatively correlated with soil Cu, Ni, Al, Fe, K, Mg, Mn, P and Ti levels (p ≤ 0.05). On the other hand, root Cr levels were found well correlated with soil Cu, Ni, Al, Fe, Mg, and Mn levels (p ≤ 0.05), where it was not correlated with these elements in soil (Laptiev et al., 2024).

Soil Na, S and Ti levels were well correlated with GT stem levels (p ≤ 0.05) (Table S17). Stem Cd levels were negatively correlated (p ≤ 0.05) with soil Al, Fe, K, Mg, Mn, Ni and P levels, and the same correlations were found for soil Cd. Stem Zn levels were also negatively correlated with soil Al, Fe, K, Mg, Mg, Mn, Ni and P levels (p ≤ 0.05), but soil Zn levels were only negatively correlated with Mn from this group (Zhou et al., 2015). Stem Ti levels were strongly correlated with soil Al, Fe, K, Mg, Mg, Mn, Ni and P, and these relationships were also valid for soil Ti levels (Hussain et al., 2021).

There were significant relationships between root and stem concentrations of Pb, Zn and Fe (p ≤ 0.05) (Table S18), indicating the transfer of these elements from the root to the above-ground parts. Cr levels in roots and stems were negatively correlated, suggesting that Cr reaches these plant parts from different sources. Cu, Zn and Mn elements, which have some correlations among themselves in the root zone, all showed statistically significant negative correlations Fe (p ≤ 0.05) with Al, Na, Mg levels measured in the stem zone. The negative and significant correlations of Cu, Zn and Mn in the root with Al, Na and Mg in the stem indicate a possible transport antagonism between these elements in the plant or a different transport profile due to stress response. This suggests physiological selectivity in the transport of elements between root and stem. In particular, the high levels of microelements such as Cu, Zn and Mn may have an inhibitory effect on the transport of macro elements such as Mg (Behtash et al., 2022; Rengel & Robinson, 1989; Wairich et al., 2022).

Source identification of elements in soil and in the parts of GT

PCA was performed to the data sets obtained from elemental analyses of soil samples and GT fragments to describe major geochemical patterns and potential pollutant sources. As a result of the analysis of the soil data, two principal components (PC) were identified, explaining 84.30% of the total variance (Table 3, Tables S19 and S20). The first component (PC1S), accounting for 70.17% of the total variance, displayed strong positive loadings for Fe, Ni, Mn, Al, K, P and Cu which are indicative of both lithogenic background and potential long-term agricultural inputs. In contrast, Pb, Cd and Zn presented moderate to high negative charges in the same component, pointing to a geochemically distinct source—possibly linked to former contamination or selective retention mechanisms in the soil.

Table 3 Rotated component matrices obtained from PCA conducted using elemental levels determined in soil, root, and stem samples.

	Soil	Root	Stem	
Elements	Component	Component	Component	
	PC1
70.17%	PC2
14.12%	PC1
32.36%	PC2
27.90%	PC3
18.92%	PC1
41.88%	PC2
35.08%	
Al	0.969		–	–	–		−0.946	
Cd	−0.654		−0.754				0.740	
Cr	–	–		−0.913		−0.827		
Cu	0.868		0.630			0.897		
Fe	0.987		–	–	–	–	–	
K	0.956		0.787			0.925		
Mn	0.972		–	–	–	0.938		
Na	–	–	0.739	0.536			−0.839	
Ni	0.975			−0.818		–	–	
Pb	−0.794				0.843	−0.685	0.602	
P	0.899		0.928			0.875		
S		0.910		0.728			−0.633	
Zn	−0.588	0.619		0.571	0.676		0.835	

The second component (PC2S) explains 14.12% of the total variance and shows high loading especially with S and Zn elements (Table 3). In line with geological data, it is known that the soils in the plain part of the study area are naturally rich in elements such as Fe, Mn, Al and K (Çevik et al., 2020; Göktaş & Hakyemez, 2015). However, various mining activities have been carried out on the slopes surrounding the plain from past to present. For example, andesite and galena were mined around Ulucak in the north, flint was mined around Yakaköy, and Pb-Zn-Cu was mined in Yukarıkızılca region in the south (Bakaç, 2000; Çevik et al., 2020; Menteşe, 2022). In this framework, it is evaluated that S and Zn elements in PC2S were transported by surface runoff and erosion processes originating from metallic mining sites in the region and accumulated in the plain soils, while Fe, Mn and K elements showing high loadings in PC1S are largely associated with andesite and flint formations (General Directorate of Mineral Research and Exploration, 2024). On the other hand, it is thought that the elements K, P and Cu with high factor loadings in PC1S may have been exogenously incorporated into the soil system as a result of long-term fertilization and pesticide use in the past and may have become the background composition of the soil over time. Although the general elemental distribution observed in the soil is largely due to the geological and geochemical characteristics of the region; it is considered that the elements such as Pb, Cd and Zn, which are difficult to be associated with the natural geochemical structure and which are included in PC1S with low but significant negative loadings, and the elements K, P and Cu may have been transported to the soil system through fertilizers and/or agricultural chemicals used in pre-industrial periods (KOSBI & Halkbankası, 2020; Rashid et al., 2023).

PCA explained 79.17% of the variance of the root element levels data set (Tables S21 & S22). The first principal component (PC1R) explains 32.36% of the total variance, where P showed the highest positive factor loading coefficient. Cu, K, Na and Cd elements were also appeared in PC1R with relatively high factor loadings (Table 3). According to the results of PCA conducted with soil data, the elements P, Cu, K and Cd co-occurred in the same principal component (PC1S). The signs of the factor loading coefficients of these elements in PC1S are also consistent with their counterparts in PC1R obtained. Therefore, it is considered that P, Cu, K and Cd levels obtained in the root zone originate from the soil. However, as discussed above, it should not be ignored that the presence of these elements in the soil is not only related to the geochemical structure but also to past agrochemical uses (Bradl, 2005; Kemalpaşa Municipality, 2024). In the second principal component (PC2R) in the root region of GT explains 27.9% of the total variance, there are two groups of oppositely charged elements, namely Cr-Ni with higher factor charges and S-Na-Zn with intermediate factor charges. Ni is known to originate from soil geochemical structure (PC1S) and there is a strong correlation between soil Cr and Ni values. However, neither Ni nor Cr levels in roots correlate with their respective soil levels. Therefore, the Ni-Cr group in PC2R is thought to be caused by anthropogenic effects. It is known that activities such as machinery, iron and steel processing, chemical and automotive industries, which constitute 60% of the industrial producers in KOSBI region, emit Ni and Cr in gas and/or liquid discharges (Table S1). Therefore, the presence of Ni-Cr in PC2R was associated with industrial emissions in the region.

As discussed above, S and Zn in PC2S are associated with the content of material transported from metallic mining sites in the region and Na, a crustal element, is expected to be present in this material. In this respect, the source of the second group of elements, S-Na-Zn, in PC2R was also considered to be material transported to the soil from mining sites. PC3R was loaded with Pb and Zn and explains 18.92% of the total variance; the relationship between Pb and Zn levels in the root zone was not statistically significant, but the correlation coefficient r: calculated as 0.526. The source of this element pair in PC3R is considered to be not only soil mineralogy but also agrochemicals used in the pre-industrial period.

In GT stem, PCA resulted with two PCs extracted 76.96% of the total variance (Tables S23 & S24). In the first principal component (PC1ST) which is explaining the 41.88% of the variance, Mn-K-Cu-P forms one group, while Cr-Pb with opposite sign forms another group of elements. The first group refers to agrochemicals originating from fertilizers, fungicides and micronutrient fertilizers in terms of the elements they contain and which have become part of the background values of the soil and can be transferred to the above-ground parts of the plant. Pb is another element that affects the elemental levels in the GT root zone and presents in the gaseous and/or liquid discharges of all the above mentioned industries widely active in the region (Table S1). However, unlike the root zone, Pb is observed in the first principal component explaining a significant portion of the total variance in the stem zone. In this respect, it is considered that Pb and the accompanying Cr observed in the PC1ST structure originate from the dominant industrial activities emitting gaseous emissions affecting the region.

In PC2ST, which explains 35.08% of the total variance, there are oppositely charged two groups of elements; Al-Na-S and Zn-Cd-Pb. Zn-Cd-Pb group represents background soil mineralogy under the influence of material transport from aforementioned mining sites, where a possible contribution of agrochemicals use is also suggested. S, which is in the second element group in the PC2ST structure together with Al and Na, which are associated with soil or soil-based materials, is the element with the highest translocation factor among those included in the component matrix and the average TF value determined in the samples was calculated as 1.2 ± 0.55. In addition, the S BCF values determined for the stem in nine of the 13 samples examined were above those determined for the root. There is no information in the literature that GT can accumulate significant levels of S in the aboveground parts. This strengthens the evidence that the S element reaches the stem region through external effects (Zeng et al., 2021). In addition to the fossil fuels used by industrial establishments, PC3 is considered the impact of fossil fuels used in the region, considering that a significant portion of the population living in Kemalpaşa district uses fossil fuels for heating purposes and has not yet switched to natural gas (Natural Gas, 2022).

Distribution of principal component scores by sampling locations

The distribution of PCA scores revealed significant differences between soil, root, and stem samples (Table S25). In soils, PC1S (70.17% variance), primarily associated with Fe, Ni, Mn, Al, K, P, and Cu elements, clearly separated areas with high positive scores (S7 and S9) from those with low scores (S1 and S3). Points S5, S8, and S13 are also moderately or weakly related to the same group of elements. It is noteworthy that all points more closely related to this group of elements are located north of the Nif Creek, which divides the site in a west-east direction. In contrast, points S1 and S3 are strongly related to Pb and Cd, which have negative loads in PC1S, and together with the negatively loaded points S6, S2, and S4, they show a more scattered distribution without a distinct spatial concentration. PC2S (14.12% variance) is associated with S and Zn, highlighting the S1, S7, S13, and S5 areas as rich in these elements and the S3, S10, S6, and S8 areas as low-contribution areas.

In root samples, PC1R (32.36% variance) revealed enrichment in P, K, and Na in RO5, RO3, RO7, and RO4 samples, while Cd accumulation was predominant in RO8 and RO11 samples. PC2R (27.9% variance) separated RO1, RO2, and RO13 samples with unique element patterns, while RO9, RO6, RO8, and RO7 samples formed the group with the lowest contribution to this component. PC3R, reflecting a separate element signature focused on Pb–Zn, explained an additional portion of the variance and revealed different groups: RO3, RO6, and RO1 showed high positive scores, while RO12 and RO13 were at the negative end.

The variance was more evenly distributed in the stem samples (PC1ST 41.88%; PC2ST 35.08%). PC1ST was associated with Mn, K, Cu, and P, with the highest scores found in the ST10, ST6, and ST9 samples and the lowest scores in the ST11, ST1, and ST13 samples. PC2ST is related to Zn, Cd, and Pb, highlighting the ST3, ST2, and ST6 samples due to the accumulation of these elements; in contrast, the ST13, ST8, and ST9 samples reflect the contribution of Al and Na elements more.

As a general assessment, PCA scores indicate significant heterogeneity among soil, root, and stem samples. The separation between the Fe-Mn-Al-K-P-Cu group and the Zn-S and Pb-Cd groups in the soil corresponds to the spatial distribution of the sites. In root and stem samples, the distribution of elements is more complex, appearing in different components. In particular, the prominence of Cd and Pb in roots and Zn in roots and stems through separate components indicates that they do not fully match the patterns observed in the soil. This situation may arise from GT’s biological uptake and transport processes, but the presence of industrial facilities in the region and the effects of wastewater and gas emissions from these facilities should also be considered. This is because variations between samples may be closely related not only to biological processes but also, and in some cases more predominantly, to the effects of multiple pollutant inputs from different sources.

The findings are consistent with the general trends in the literature regarding the transfer of heavy metals from soil to plants. However, the unique environmental and land use history of the Kemalpaşa region has led to the emergence of different distribution patterns for some elements. In particular, the accumulation patterns observed in GT have not been previously reported in the literature. This highlights the novel perspective and contribution of the study. By combining compositional data analysis with PCA, this study provides a more detailed framework on element interactions and potential pollution sources, thereby expanding the existing knowledge and filling a gap in this field. However, we express this emphasis on contribution in a more balanced manner, considering that emissions from Kemalpaşa’s industrial activities and different pollution sources may also influence these unique patterns. In this way, our study addresses both regional environmental dynamics and general trends in the existing literature within a broader context, offering a balanced and comprehensive contribution to the literature.

The results of human health risk assessment

Nemerow composite pollution index

The Nemerow Composite Contamination Index (NCPI) is a widely used mathematical method to assess the integrated contaminant load when more than one contaminant is present in foods. In this study, NCPI values for edible parts of GT plant were calculated and presented in Fig. 3, Tables S26 and S27. All single factor indices (Pi) for Cu, Pb, Zn, Fe and Mn were >1.0, indicating that the samples may be contaminated with these metals.

In most of the samples, root tissues had higher contamination levels than stems. In RO3, RO4, RO5, RO5, RO6, RO6, RO7, RO7, RO8, RO8, RO10, RO11 and RO12, NCPI values were in the range of 2.5–7, indicating moderate contamination, and in RO1, RO2, RO9 and RO13, values >7, indicating high contamination. In the stem samples, ST3, ST6, ST7, ST8, ST8, ST10, ST11 and ST12 were slightly contaminated, ST2, ST4, ST5, ST9 and ST13 were moderately contaminated and ST1 was considered highly contaminated.

In particular, the sample at point S1 had the highest NCPI value in edible tissues, indicating the general level of contamination. Although NCPI is widely used in the assessment of health risks in different foods, it was applied for the first time for GT plant in this study. In a study conducted by Román-Ochoa et al. (2021) in Peru, NCPI values were found to be in the range of 0.07−0.28 in cereals from farmland far from contamination sources, indicating a low risk level. In contrast, in a study on vegetables by Proshad & Idris (2023), NCPI values ranged from 7.92–14.18; stem tissues had higher values compared to stem samples in this study, while the situation was reversed in root samples.

Furthermore, a very strong correlation was found between stem and root NCPI values when all sample points were considered (r = 0.955, p ≤ 0.001). The correlation remains statistically significant even when sample S1 is excluded from the data set (r = 0.627, p ≤ 0.05), indicating the consistency of contamination levels between the two tissue types.

Estimated daily intake of heavy metals

The daily intake (EDI) of heavy metals of individuals was evaluated according to their dietary requirements. None of the EDI values calculated for stems or roots exceeded TUILs for adults (Tables S28 and S29). On the other hand, at all sampling points, EDI values for Cu in stems and roots and Mn in roots were above the TUIL value specified for children (Fig. 4, Tables S30 and S31). In addition to that, the EDI values calculated for Mn in stem samples exceeded the TUIL for children with exception for samples ST8 and ST11. EDI values calculated for Fe (in RO1, RO2, RO9 and RO13) and Cr (in RO6, RO7, RO8, RO9, RO10, RO11, RO12, RO13, ST1, ST2, ST3, ST4, ST4, ST5, ST11 and ST13) in some sampling points also exceeded TUIL for children. Moreover, EDI values for Zn in RO1, RO2, RO3, RO4 and RO6 exceeded TUIL. None of the EDI values calculated for Cd, Ni, and Pb exceeded the TUIL value for children in any of the sampling points and plant parts.

Figure 3 NCPI values for (A) root and (B) stem samples.

Figure 4 EDI values calculated for children in GT.

sections (A) Cr, (B) Cu, (C) Fe, (D) Mn and (E) Zn (mg/kgbw day).

Only the publication by Jalali & Fakhri (2021) provided EDI data by analyzing Cd, Cu, Fe, Mn, Ni, and Zn in GT plants collected from rural areas and local markets in western Iran. Researchers have reported EDI values ranging from 0 to 0.0009 mg/kgbw day for adults and 0 to 0.0013 mg/kgbw day for children; these values are significantly lower than the results obtained here. This situation reveals the extent of heavy metal accumulation in the study area in Kemalpaşa.

Noncarcinogenic health risk assessment

The non-carcinogenic risk assessment was conducted in accordance with the standards of the United States Environmental Protection Agency (USEPA, 2002). Hazard quotient (HQ) and hazard index (HI) values for root samples for adults and children are presented in Tables S32 and S33. For adults, the HQ values for Cu in all root samples were above 1; indicating a potential health risk. In general, HI ≥ 1 values express a potential non-cancer risk for adults. Additionally, in the assessment conducted for children, the elements Cr, Cu, Fe, Mn, Pb, and Zn pose potential health risks. The high HI values obtained for children (85.0–138.97) indicate that non-cancer effects may be more significant. HQ and HI values for stem samples are presented in Tables S34 and S35. In adults, the Cu level at ST5 poses a potential health risk (HQ ≥ 1), and accordingly, HI ≥ 1 values indicate the potential of non-cancer risks. ST2, ST4, ST5, ST6, and ST10 samples are found to carry a potential non-cancer risk for adults. For children, Cu and Mn HQ values were determined to be >1 in all samples, while the same was the case for Zn and Fe in some samples. Overall, it is understood that there is a potential non-cancer risk for children due to HI ≥ 1 values in all plants’ parts.

In a study conducted on GT plants in Iran, Cd, Cu, Fe, Mn, Ni, and Zn elements were analyzed, and HQ values ranging from 0.0004 to 0.0458 for adults and 0.0008 to 0.0985 for children were reported (Jalali & Fakhri, 2021). These values are significantly lower than the HQ values obtained in this study, especially for children.

Carcinogenic risk assessment

The lifetime cancer risk (CR) values calculated for adults and children for the elements Cd, Cr, Ni, and Pb, which are classified as carcinogenic by the International Agency for Research on Cancer (International Agency for Researchon Cancer IARC (2025)), are presented in Tables S36 and S37. For adults, CR values of the corresponding elements were within acceptable limits (1 ×10−6–1 ×10−4). For children, Cr concentrations in root samples indicated a high cancer risk, while Pb concentrations were generally within acceptable limits. For Ni, the risk in the RO9, RO10, and RO12 samples was unacceptable for children. While the RO3, RO4, RO5, and RO7 samples remained within safe limits for Cd, CR values for other samples fell into the unacceptable/high-risk category. Overall, CR values for children are high, and there is a significant risk associated with carcinogenic metals.

A similar assessment was conducted for stem samples, and the results are presented in Tables S38 and S39. For adults, Cr and Ni risk levels were mostly within acceptable limits, and in some samples, Ni risk was also negligible (CR <1 ×10−6). Additionally, Pb risk levels were also negligible. However, CR values for Cd were in the unacceptable category for ST1, ST2, and ST3. This indicates the presence of Cd-related cancer risk for adults regarding stem samples. For children, CR values for Cd and Cr were in the unacceptable class. While risk levels for Ni and Pb were mostly within the acceptable range for children, unacceptable risk for Ni was observed in some samples. Overall, it was concluded that the total carcinogenic risk was quite high for all samples studied.

Conclusions

Here, a comprehensive assessment of the effects of natural and anthropogenic elemental sources on the soil and the edible wild plant Gundelia tournefortii (GT) in Kemalpaşa, Türkiye, where has a history of agricultural use and has been home to intensive industrial activity for many years, is presented.

The results indicated multi-element contamination, particularly with very high enrichment of Pb and widespread to moderate enrichment of Ni in the soil. Significant accumulation of certain metals in some GT plants and of nutrients at all points revealed strong soil-plant transfer dynamics. The wide distribution of bioconcentration factors in roots and stems suggested geochemical variability in the area and multiple sources of contamination. Elements that show low enrichment in the soil but high accumulation in the plant underlined the possible effects of atmospheric and/or industrial inputs.

PCA supported these observations by revealing distinct patterns of elemental origin. PCA conducted with soil analysis data showed that the elements found in the region’s geochemical structure and the elements representing the agrochemicals used in its long agricultural history are intertwined in a way that cannot be fully separated from each other as clusters and that they together constitute the soil background structure. The effects of this condition observed in the soils were also reflected in GT parts. While PCA does not clearly reveal evidence of industrial pollution in soil, the impact of industrial pollutants is observed to account for 27.9% of the total variance in the plant root zone data set. In the stem part, the effect of industrial emissions is observed in a principal component explaining 41.88% of the variance, while the effect of fossil fuels used in households and industry stands out in another principal component explaining 35.08% of the variance.

The study revealed concerning findings related to public health. Health risk assessments showed that root tissues exhibited higher contamination levels than stems in most samples. Furthermore, these assessments have indicated that GT consumption may have short- and long-term effects on children.

The results show that, despite the strong influence of geogenic traces in the region, anthropogenic pressures such as agriculture, industrial emissions, and fossil fuel combustion are clearly detectable in the investigated matrices. The findings also indicate that ecotoxicological risks and potential health risks in terms of food safety may vary depending on the part of the plant consumed. For example, the accumulation of toxic elements such as Pb and Cd in the roots and their transfer to the stem indicate a significant risk, which is a situation that should be considered in terms of consumption. Therefore, the results indicate that more detailed risk assessments should be conducted in areas where agricultural production and industrial activities interact. Furthermore, the differences between the results obtained from spatially distinct points also provide a scientific basis for land use planning, industrial site selection, and agricultural management practices. Spatial planning systems and related institutions play an important role in managing non-agricultural land expansion, and comprehensively assessing the impacts that non-agricultural applications will have on agricultural land is of great importance for improving environmental and health policies. Applying more strict limitations on the amount of agricultural land used for urban or industrial development should be an international policy priority aimed at preserving the quantity and quality of existing land resources for safe food production and sustainable development.

The study highlights the need to: (i) conduct a more in-depth investigation into the potential health risks associated with the inclusion of WEPs collected from nature in the diet, particularly in complex areas like Kemalpaşa where industrial and residential zones are intermingled with former agricultural lands, (ii) the preference for areas free from the effects of polluting sources for the cultivation of wild plants and agricultural products that are traditionally and widely consumed, and (iii) implement emission reduction and monitoring strategies appropriate to the land use structure in the region in order to protect food safety and ecosystem health.

Supplemental Information

Supplemental Information 1 The main heavy metals and other elements found in industrial emissions

Supplemental Information 2 The heavy metal values in soils, crops and other WEPS in previous studies

Supplemental Information 3 Heavy metals and other elements concentrations in soil samples

Supplemental Information 4 Heavy metal levels recorded in soil and GT in previous studies (mg/kgdwa )

Supplemental Information 5 The detection limits of XRF and ICP-MS, mg/kgdw

Supplemental Information 6 pH, EC, and organic matter levels of soil samples

Supplemental Information 7 The pollution index values (PI)

Supplemental Information 8 Heavy metals and other elements concentrations in root samples

Supplemental Information 9 Heavy metals and other elements concentrations in stem samples

Supplemental Information 10 Heavy metals and other elements concentrations in root samples in fresh weight

Supplemental Information 11 Heavy metals and other elements concentrations in stem samples in fresh weight

Supplemental Information 12 Bio-concentration and translocation factors of heavy metals and other elements

Supplemental Information 13 Correlations among the levels of heavy metals and other elements in soil samples

Supplemental Information 14 Correlations among the levels of heavy metals and other elements in root samples

Supplemental Information 15 Correlations among the levels of heavy metals and other elements in stem samples

Supplemental Information 16 Correlations among the levels of heavy metals and other elements in soil and root samples

Supplemental Information 17 Correlations among the levels of heavy metals and other elements in soil and stem samples

Supplemental Information 18 Correlations among the levels of heavy metals and other elements in root and stem samples

Supplemental Information 19 Total variance explained by PCA for soil samples

Supplemental Information 20 KMO and Bartlett’s test results for soil samples data set

Supplemental Information 21 Total variance explained by PCA for root samples

Supplemental Information 22 KMO and Bartlett’s test results for root samples data set

Supplemental Information 23 Total variance explained by PCA for stem samples

Supplemental Information 24 KMO and Bartlett’s test results for stem samples data set

Supplemental Information 25 Distribution of principal component scores by sampling locations

Supplemental Information 26 The Nemerow Composite Pollution Index (NCPI) values for roots

Supplemental Information 27 The Nemerow Composite Pollution Index (NCPI) values for stems

Supplemental Information 28 Estimated daily intake (EDI) of heavy metals in root samples for adults

Supplemental Information 29 Estimated daily intake (EDI) of heavy metals in stem samples for adults

Supplemental Information 30 Estimated daily intake (EDI) of heavy metals in root samples for children

Supplemental Information 31 Estimated daily intake (EDI) of heavy metals in stem samples for children

Supplemental Information 32 Hazard quotient (HQ) and hazard index (HI) of heavy metals in root samples for adults

Supplemental Information 33 Hazard quotient (HQ) and hazard index (HI) of heavy metals in root samples for children

Supplemental Information 34 Hazard quotient (HQ) and hazard index (HI) of heavy metals in stem samples for adults

Supplemental Information 35 Hazard quotient (HQ) and hazard index (HI) of heavy metals in stem samples for children

Supplemental Information 36 Lifetime Cancer Risk (CR) of heavy metals in root samples for adults

Supplemental Information 37 Lifetime Cancer Risk (CR) of heavy metals in root samples for children

Supplemental Information 38 Lifetime Cancer Risk (CR) of heavy metals in stem samples for adults

Supplemental Information 39 Lifetime Cancer Risk (CR) of heavy metals in stem samples for children

Supplemental Information 40 Sampling points and automative industries

Supplemental Information 41 Sampling points and chemical industries

Supplemental Information 42 Sampling points and industries of construction materials

Supplemental Information 43 Sampling points and electric-electronics industries

Supplemental Information 44 Sampling points and food industries

Supplemental Information 45 Sampling points and iron and steel processing industries

Supplemental Information 46 Sampling points and machine production industries

Supplemental Information 47 Sampling points and paper and paperboard industries

Supplemental Information 48 Sampling points and plastic industries

We would like to express our sincere thanks to the Solid Waste and Soil Pollution Laboratory (SoLAB) and the Science and Technology Application and Research Centre (TEAM) of Dokuz Eylül University (DEU), where the experimental studies were carried out.

Additional Information and Declarations

Competing Interests

Author Contributions

Data Availability

The authors declare there are no competing interests.

Ayşenur Özuysal conceived and designed the experiments, performed the experiments, analyzed the data, prepared figures and/or tables, authored or reviewed drafts of the article, and approved the final draft.

Fariborz Fadaeivash performed the experiments, prepared figures and/or tables, and approved the final draft.

Görkem Akıncı conceived and designed the experiments, analyzed the data, prepared figures and/or tables, authored or reviewed drafts of the article, and approved the final draft.

The following information was supplied regarding data availability:

The raw measurements are available in the Supplementary Files.

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
