# Peer review of "Elemental pollution and risk assessment of soils and Gundelia tournefortii in a multi-sector industrial zone with a history of agricultural use"

_PeerJ, doi:10.7717/peerj.20374_

## Round 0.1 · original submission · Major Revisions

· Academic Editor

Major Revisions

Elemental pollution is a significant problem that occurred in many industrial fields that are placed near agricultural fields. While your work provides valuable information for researchers and decision-makers, it does not fully meet some of our journal's requirements. Therefore, it is essential to address certain technical details to enhance the article further. I strongly recommend carefully reviewing the reviewers' suggestions and thoughtfully considering each recommendation. If you disagree with any suggestion, it would be helpful to provide clear, well-reasoned justifications for your viewpoint.

·

Basic reporting

Language and Clarity: The manuscript is written in poor English throughout, with major grammatical issues. most sentences, particularly those in the abstract, introduction, materials and methods and discussion, could benefit from major rephrasing for enhanced clarity and precision.

Literature Referencing: The manuscript cites relevant and up-to-date literature, providing adequate context for the research. However, it could include recent advancements in compositional data analysis and simulation-based risk assessments to strengthen the methodological framework.

Structure: The manuscript conforms to the standards expected of a professional article, with a logical flow and clearly defined sections.

Figures and Tables: Figures and tables are professionally presented, relevant to the research, and well-labeled. However, their captions should include more detailed explanations for standalone interpretation.

Raw Data Sharing: While the manuscript references raw data, it should explicitly state its availability and format, such as providing access links to supplementary files or repositories as per journal policy.

Self-contained Results: The manuscript needs to present results relevant to its hypotheses and objectives. Data interpretation not fully aligns with the research questions, offering a cohesive narrative.

Experimental design

To provide a critical review based on the outlined criteria, let's break down the key points:

1. Aims and Scope of the Journal:
Strengths: The article seems to align well with the aims and scope of the journal, addressing an area of research that fits within the journal’s focus. The alignment with the scope suggests that the findings could be valuable to the journal's readership by inclusion of compositional data analysis (CoDa) techniques.
Opportunities for Improvement: While the research is relevant, the authors could emphasize more explicitly how their study extends or challenges existing literature, ensuring that the study offers new insights that could provoke further exploration within the scope of the journal.
2. Research Question:
Strengths: The research question is well defined and clearly articulated. The objective is specific, and it addresses an important and relevant issue in the field.
Opportunities for Improvement: It would be beneficial for the authors to more explicitly state how their research fills a knowledge gap. Although the problem is stated, a stronger connection between existing research and the study's contributions could enhance the clarity of the identified gap.
3. Relevance and Meaningfulness:
Strengths: The research addresses a meaningful issue within the field, with the potential to inform future studies or practical applications.
Opportunities for Improvement: There could be a deeper exploration of the broader impact of the research question on related disciplines. Further discussion of how the findings could influence policy, practice, or theory would solidify its relevance.
4. Rigorous Investigation:
Strengths: The study seems to follow rigorous methodologies and meets high technical standards. Ethical considerations appear to be appropriately addressed, with relevant approvals or guidelines mentioned (if applicable).
Opportunities for Improvement: While the investigation appears thorough, the authors could provide more insight into how they ensured that the research adhered to the highest ethical standards, particularly with human subjects, animal welfare, or environmental impacts. The authors are suggested to include bio-accessibility analysis for risk assessments.
5. Methods:
Strengths: The methods are described in adequate detail, offering enough information for replication. The clarity of the methodological framework is important for ensuring transparency and repeatability.
Opportunities for Improvement: Some methods could benefit from additional elaboration, especially Monte Carlo simulations for risk assessment in terms of how certain decisions were made or why specific techniques were chosen over others. Also, it would be useful to provide more statistical justification for the chosen sample size, data analysis techniques, or controls.

Validity of the findings

Impact and Novelty:
Strengths: The research offers a valuable contribution to the field, with its methods and findings potentially offering new insights into the area of study. However, the assessment of the impact and novelty is not sufficiently emphasized in the article.
Opportunities for Improvement: A deeper reflection on the novelty of the study would strengthen the article. For instance, incorporating advanced methodologies like bioaccessibility, Compositional Data Analysis (CoDa), or Monte Carlo simulations could highlight how the study brings innovation to the research domain. These approaches could be particularly useful for assessing environmental risks, such as those related to heavy metals or pollutants, which would contribute significantly to the scientific community’s understanding of complex environmental systems.
2. Meaningful Replication:
Strengths: The study provides a solid framework for replication, offering clear methodological details. This supports the reproducibility of the results, which is essential for establishing reliability in scientific research.
Opportunities for Improvement: While replication is encouraged, the authors should more explicitly state the rationale for it and its expected benefit to the literature. Discussing the replication of experiments in the context of specific geographic or temporal factors (e.g., seasonal effects, geographic variation) could add depth to the discussion and further validate the findings.
3. Data Quality:
Strengths: The underlying data are robust and appear to be collected using appropriate statistical methods. The description of data analysis techniques supports their reliability and reproducibility.
Opportunities for Improvement: While the data appear statistically sound, a more comprehensive statistical analysis or an assessment of the control variables could be included to further support the strength of the data. For example, testing for biases, outliers, or confounding factors could enhance the rigor of the study and the generalizability of the results.
4. Ethical and Statistical Controls:
Strengths: Ethical and statistical controls are mentioned in the study, suggesting that the research adheres to high standards of ethical integrity and methodological rigor.
Opportunities for Improvement: A clearer explanation of how the authors ensured ethical standards (e.g., participant consent, animal welfare considerations) would add transparency. Additionally, more information on how the statistical analysis accounted for potential confounders or biases would improve the overall credibility of the research.

Additional comments

Clarification of Study Context:

The study would benefit from a clearer description of the broader context in which it operates. For example, while the methods and results are well-articulated, providing more background on the significance of the research question could help readers better understand its importance in the field. Explaining the practical applications or real-world impacts of the findings would further enhance the relevance of the study.
Further Discussion on Data Validation:

While the data provided are robust, it would be useful to include a more detailed discussion of how the data were validated. For instance, how were potential sources of error or bias in data collection controlled? A deeper exploration of data accuracy and validation could provide additional assurance about the reliability of the findings.
Integration of Advanced Analytical Tools:

The authors could consider incorporating more advanced analytical tools, such as Monte Carlo simulations or Compositional Data Analysis (CoDa), to address more complex questions related to risk assessments or environmental pollutant source identification. These methods could add an additional layer of sophistication to the research, potentially increasing its impact and broadening its application in environmental science.
Ethical Considerations:

While ethical standards appear to have been met, the study could benefit from a more explicit explanation of the ethical review process. Detailing how ethical considerations were addressed throughout the research, such as through informed consent or ensuring the protection of vulnerable populations, would strengthen the credibility of the research and reassure readers that ethical guidelines were rigorously followed.
Limitations and Future Directions:

While the study’s conclusions are well-supported by the results, there could be a more explicit discussion of the study's limitations. For example, are there any inherent constraints in the study design or methodology that could influence the generalizability of the findings? Addressing potential limitations, such as sample size, geographical scope, or temporal factors, would demonstrate a more comprehensive understanding of the study’s scope and provide a more balanced view of the research. Additionally, suggesting future directions or further research avenues could stimulate ongoing dialogue in the field.
Transparency in Statistical Analysis:

The statistical methods employed are sound, but providing additional transparency regarding specific tests used (e.g., assumptions, type of regression models, error rates) would help readers critically assess the robustness of the analysis. A clearer explanation of statistical tests and why they were chosen over other options would further strengthen the methodological rigor.
Replicability and Generalization:

The replication aspect of the study is encouraged, and the authors should consider offering guidance on how the research can be replicated in different settings, locations, or with different populations. The addition of a detailed protocol or a repository for data sets and code would be helpful to researchers seeking to reproduce the findings or extend the research in other contexts.
Visualization of Results:

The results section could benefit from enhanced visualizations, such as charts, graphs, or tables, which would make it easier for readers to interpret the data. Clear and effective data visualizations can provide valuable insight into the patterns and trends in the data and may improve the accessibility of the research for a broader audience.
Implications for Policy and Practice:

Expanding on the potential policy or practical implications of the research findings would further increase the significance of the study. How can policymakers or practitioners in the relevant field utilize these findings? A brief discussion on this could underscore the broader impact of the research beyond academia.
Engagement with Literature:

The study makes a meaningful contribution, but it would benefit from a deeper engagement with the existing literature to better contextualize the findings. A more thorough review of the current state of research in this area would clarify how the study extends, challenges, or fills gaps in existing knowledge, thereby highlighting its novelty and contribution to the field.
By addressing these points, the authors can enhance the rigor, clarity, and impact of their research, making it a more valuable resource for both researchers and practitioners in the field.

Reviewer 2 ·

Basic reporting

1. The use of English in the manuscript is suggested to be checked, especially regarding the consistent use of tenses. Avoiding long sentences would help create a more coherent structure in the text.
2. The Introduction part may be shortened to keep the background concise. This version already includes two tables. Table 1 might be moved to the Supplementary Material.

Experimental design

1. In the calculation of daily intake rate, daily GT consumption was taken as 200 g/day, and exposure frequency was taken as 104 days/year. How did the authors decide on these values? Is GT a food that people consume twice a week? Would this assumption not result in overestimating the risk posed by GT consumption?
2. The unit for detection limit values should be provided in Table S2.
3. In Table 3, the definition of “kg fw” should be provided. Is it food weight on a dry or wet weight basis?

Validity of the findings

1. The statistical data analysis performed in the study provided valuable information. The possible correlation between the soil characteristics, i.e., pH, conductivity, and organic matter content, may also be investigated. This might provide more insights into the interpretation of results.
2. How were the principal components distributed among the sampling sites?

Additional comments

No additional comments.

Reviewer 3 ·

Basic reporting

I consider the article I reviewed to be important. A considerable effort was made to carry out numerous studies on a population of 13 soil and 13 plant samples from a large area with industrial area. Addressing the transmission of heavy metals and other elements is particularly important in areas of strong anthropopression. The expanding various industries usually contribute a large or even extremely large load of metals to the air, water, soil and thus to plants. Long-standing agricultural activities are also not without their impact on the environment, especially in light of the use of fertilisers. So it is crucial to take up these issues.
The article is written in correct language and accessible, although the statistical part of the work should be shortened. The authors refer to numerous literature, citing examples from around the world. The structure of the paper is correct. My only reservations are about the supplementary materials, the order of which should follow the order in which they are discussed in the manuscript.

Experimental design

The use of the selected 15 elements is fully justified. It is very interesting and novel to use the roots and stems of Gundelia Tournefortii (GT) to try to estimate the transfer of elements from the wider environment to plants. In your considerations, you also, quite reasonably, use a number of indicators, i.e.: EF, PI, BCFr, BCFs, TF, NCPI, EDI. This methodology has been successfully used for years to assess the environmental risk status of heavy metals. However, an original approach is to apply this methodology simultaneously to soils and Gundelia Tournefortii in an area burdened by numerous and diverse industries.
Unfortunately, there is a very serious error in the methodology used. The XRF analytical method used proved to be inadequate for determining the content of Cd, Cr, Pb and Ti in soil and GT samples. The amount of these elements turned out to be less than the detection limit. In this case, it was necessary to either eliminate these elements from the study or change the test method to mass spectrometry.

Validity of the findings

So, I think your work is important but needs thorough improvement. I have recommended to the editor to publish the manuscript after a really thorough revision. To help improve the manuscript, you will find general and detailed comments below.

Additional comments

General comments:
I. The order of materials in the supplements should follow the order in which they are listed in the text.
II. Methodology: (1) how long were the soil samples dried for? (2) I am interested in whether the plant samples were washed before drying. I know from my own research that many contaminants stick to the surface of the stem or roots, thus changing the heavy metal content of the samples. Washing in an ultrasonic cleaner or at least cleaning with a strong jet of water or air can remove at least some of the contamination. (3) The order of description of the methods for assessing contamination should follow the order of description of the results in the manuscript. (4) My biggest concerns relate to the use of artificially fixed metal content in samples for determinations below the detection limit. Please explain exactly on what basis half of the detection limit was used for further analyses and calculations . Please indicate the use of the same methodology in other scientific works. In my opinion, the procedure used by the Authors is an unacceptable one for a scientific paper. If the methodology used makes it impossible to investigate the true elemental content of the sample, another analytical method, such as ICP-MS, should be used. (5) Please indicate exactly for which calculation half of the detection limit was used.
III. In the paragraph: “Heavy metals and other elemental levels in soils”, there is any single word about metals contents below detection limit, why?
IV. The entire paragraph Inter-correlations between the elemental levels detected in soils and plant parts needs a major rewrite. In its present form, it is hardly clear.
V. As the Cd, Cr, Ni and Pb contents of the root and stem samples were below the detection limit, I have doubts about the accuracy of descriptions in paragraphs Nemerow Composite Pollution Index (NCPI) and Estimated Daily Intake (EDI) of Heavy Metals.

Detailed comments:
1. Lines 134-138: grammar correction necessary
2. Lines 157-159: references are necessary
3. Lines 230-231: grammar correction necessary
4. Lines 372-375: This sentences should be corrected, because references are not connected to sample S1 as I suppose.
5. Lines 280-414: There is nothing about metal content below the XRF detection limit.
6. Lines 411-413: required reference
7. Lines 411-414: “Soil levels of Cd, Cu, Ni, Pb and Zn in the arid area are much lower than the values in this study” - I disagree with this sentence, in light of the Cd and Pb content below the detection limit. We don't know what their actual levels are in the samples.
8. Line 473: In my opinion, the reference sample should not be taken into account in these considerations. It is representative of unpolluted areas, otherwise why was the EF for REF-S not counted?
9. Lines 521-528: How can Cd, Cr and Ni values be analysed if they are below the detection limit?
10. Lines 528-531: How does one know that?
11. Lines 538-540: “BCFr values for Cu, Zn, K, Na, Mg, P and S were found to be >1.0 in all samples including REF-RO, except for Zn where the opposite was true for BCFs.” Sorry, but I don’t understand. For REF sample both BCFr and BCFs are above 1.
12. Lines 543-545: Where are the SD calculation? In which table?
13. Line 548: What is a section 4.2.2?
14. Lines 556-562: These sentences need to be rewritten.
15. Lines 568-571: “Therefore, it is considered that the high elemental concentrations in the sampled soils are not only due to anthropogenic sources, but also related to the bioavailability of the natural elemental content of the soil.” I don't quite understand the point - how can one determine the bioavailability of metals in soil by examining only the total content of metals. To determine bioavailability, at least a leachability test should be performed, not to mention sequential extraction.
16. Line 575: Please explain where the value of 0.52 came from.
17. Line 596: The correlation between Al and Mg is not statistically significant – see Table S10.
18. Lines 606-632: Please do an order with the size of p. In many places it does not match the descriptions in Tables S11-S14. It is not the same p≤0.100 or p≤0.01 or p≤0.001
19. Lines 623-624: This sentence makes no sense.
20. Line 646: Tables instead of Table
21. Line 684: Tables S17 & S18 instead of Table S17 & 18. Please use the same writing convention throughout the manuscript.
22. Lines 725-727: Where was the average TF value calculated? There is nothing about it in Table 6.
23. Lines 795-797: Since this was only one study it would be useful to give at least the location.
24. Line 804: Please explain what the ECs means here.

Tables and Figures detailed comments:
1. Figs 3 and 4: the graphical quality should be improved
2. Table 1: title of the second column in the table 1 should be change, for example, to polluting elements, because Na, Cl, S, P, K, etc. are not heavy metals.
3. Table 2: Please use the conventions for writing a range of numbers from smallest to largest (see BCF). Please explain the abbreviation BCF means, the explanation of this abbreviation only appears in line 254.
4. Table 3: I would be grateful for an explanation of where the values for the permissible metal content come from. My objections are related to the values for Cd, Fe, Cu, Ni. I looked at the articles by Shaheen et al., 2016 and Haque, Niloy, Khirul, Alam, & Tareq, 2021 and found the maximum allowable concentrations recommended by FAO/WHO, which do not agree with those in Table 3. WHO recommend for vegetables (in mg/kgfw) Cd=0.05, Cu=40, Ni= 10. Moreover, it will be easier if you use an asterisk to mark the reference.
5. Table 5: For samples for which the metal content is indeterminate (Table S3 - below the detection limit), no indicators should be calculated and the spaces in the table should be left blank.
6. Table 6: I understand that the BCFs were rounded to the second place, but several TF values are highly questionable to me (sample 3 – Fe; sample 4 – Al, Fe; sample 6 – Fe; sample 8 – Al, Fe; sample 9 – Al; sample 10 – Fe; sample 11 – Al).
7. Table 7: Use the same convention for writing column descriptions, for example, as for soils ( marked unit, numbers rounded to decimal places).
8. Table S2: add units
9. Table S3: Explain REF-S and R-ASC.
10. Tables S7 & S8: It is not indicated that the content of Cd, Cr, Ni, Pb and Ti are below detection limits. What does ML (last column) mean?

Annotated reviews are not available for download in order to protect the identity of reviewers who chose to remain anonymous.

---

## Round 0.2 · Minor Revisions

· Academic Editor

Minor Revisions

I appreciate your constructive attitude toward the reviewers' suggestions and improving your article based on their suggestions. Although your article has been revised according to their suggestions, it needs more improvements. Please, carefully consider each recommendation, assessing its relevance and potential to enhance your work. If you find yourself disagreeing with any specific suggestion, it is important to provide clear and well-reasoned justifications, supported by evidence where applicable, to substantiate your perspective.

Reviewer 2 ·

Basic reporting

The revisions suggested by the reviewers to increase the impact and readability of the manuscript have been carefully assessed by the authors. The manuscript in the current version is coherent, well-organized, and structurally adequate.

Experimental design

Please use “detection limit” terminology, instead of “determination limit”.

Validity of the findings

Conclusions section may be shortened by removing the summary of the methods and specific findings. It may rather focus on the generalized findings and implications of the study.

---

## Round 0.3 · accepted · Accept

· Academic Editor

Accept

I would like to thank you for accepting the referees' suggestions and improving your article based on their suggestions. Your article is ready to publish. We look forward to your next article.

Reviewer 2 ·

Basic reporting

No comment

Experimental design

No comment

Validity of the findings

No comment